# Exploring the Factors Influencing the Acceptance of IoT Applications in Food Packaging

**DOI:** 10.3390/foods14040575

**Published:** 2025-02-10

**Authors:** Konstantinos Rotsios, Dimitris Folinas, Chrysoula Mouchtari, Artemis Andreou, Thomas Fotiadis, Maria-Theodora Folina, Antonios Gasteratos

**Affiliations:** 1Business Division, American College of Thessaloniki, 54631 Thessaloniki, Greece; krotsios@act.edu; 2Department of Supply Chain Management, International Hellenic University, 60100 Katerini, Greece; 3Department of Production and Management Engineering, Democritus University of Thrace, 67100 Xanthi, Greece; crisoulapersonal@yahoo.gr (C.M.); artemisandreou@gmail.com (A.A.); dr.fotiadis.thomas@gmail.com (T.F.); agaster@pme.duth.gr (A.G.); 4Department of Applied Informatics, University of Macedonia, 54631 Thessaloniki, Greece; ics22059@uom.edu.gr

**Keywords:** internet of things (IoT), food value chain, packaging, sustainability, circular economy

## Abstract

This study explores the transformative role of IoT in enhancing efficiency across the food value chain. It combines secondary and primary research to identify factors that influence IoT adoption in the food sector. A thematic–bibliometric analysis (2010–2024) highlights key themes related to IoT use in the industry, particularly in food packaging and supply chain optimization. The primary research, based on 162 questionnaires, examines factors influencing IoT adoption, such as perceived usefulness, ease of use, self-efficacy, and personal innovativeness. Results show that IoT awareness strengthens the impact of personal innovativeness on attitudes toward IoT. Additionally, a positive attitude significantly influences the intention to use IoT. These findings underline IoT’s potential to enhance efficiency, reduce waste, and positively impact the food industry, emphasizing the importance of increasing awareness to foster its adoption.

## 1. Introduction

In today’s dynamic and highly competitive business environment, consumers are becoming more and more sensitive to environmental and climate issues [1]. In addition, environmental concerns lead consumers to purchase more sustainable products, including food products [2]. The sustainability of food systems has become one of the most important challenges of our time, given environmental pressures and the need for better management of natural resources [3]. Food packaging serves the food industry and the consumers in several ways, as it preserves food quality and ensures food safety; however, its negative impact on the environment cannot be underestimated. 

The negative consequences of food packaging are apparent through the whole supply chain and can occur from the production and processing of food-to-food waste due to packaging at the retail and consumer level all the way to packaging recycling and landfill disposal [4]. There are several developments in food packaging resulting from growing consumer concerns about the environment. 

In recent years, efforts to improve packaging performance to reduce waste and enhance recyclability have increased [5]. The sustainability of food packaging can be increased in three different phases, specified as follows. (1) At the raw materials level, the use of recycled materials and of renewable resources are two strategies for reducing CO_2_ emission and the recourse to fossil resources; (2) at the production level, it can be made sustainable through more energy-efficient processes; and (3) at the waste management level, we can consider reuse, recycling, and biodegradation [6]. 

On one hand, much effort has been dedicated to decreasing the impact of packaging through the development of novel bio-based materials, the optimization of packaging use, and the improvement of materials’ performance which, in turn, allows a shift to lighter and thinner packages. On the other hand, packaging innovations have been developed, with the aim of increasing the packaged product quality, extending the shelf life, and ultimately reducing the possibility of food turning into waste [7]. The integration of biodegradable materials, recyclable packaging, and IoT technologies in the food industry is making a significant contribution to reducing environmental impacts, improving sustainability, and providing efficient solutions in food systems [8,9]. 

Overall, it appears that use of IT can play a critical role in addressing the challenges faced by the food packaging industry [10]. However, the factors that influence the application of IoT in the food and beverage sector remain underexplored [11]. Furthermore, the rapid developments in the field and the accumulated knowledge call for an updated investigation [12]. Some studies have analyzed the importance of sustainable packaging technologies and the integration of IoT in waste management. However, the integration of IoT technologies with returnable recycling solutions is underexplored, leaving a knowledge gap in the literature [13,14]. This study aims to fill this gap. More specifically, it aims to (a) identify the patterns, trends, and impact of IoT use in food packaging and (b) explore attitudes and the intentions of industry representatives regarding the future use of IoT on food packaging. Only a limited number of researchers have integrated attitude and intention into one model to better comprehend the factors that impact intention and are mediated through attitude [15]. 

The study herein contains two separate studies: (a) a thematic–bibliometric analysis on IoT and food packaging and (b) a quantitative analysis on the factors that affect the behavioral intention around Internet of Things (IoT), concerning its possible future use in food packaging.

This study is structured as follows. First, a literature review of new packaging technologies and IoT applications in the food industry is presented. Moreover, to strengthen the above review, a bibliometric analysis on IoT and food packaging is conducted. In the following Section 2, the main parameters of the research methodology are provided, and the hypotheses are developed. The presentation and analysis of the research are included in Section 3. Lastly, the conclusions of the study are discussed, and suggestions for future research are presented.

## 2. Literature Review

### 2.1. Integration of IoT Technologies in Food Packaging

The integration of IoT technologies in food packaging presents new opportunities to improve the sustainability and efficiency of the supply chain. Smart packaging, containing IoΤ sensors, provides continuous information on the status of products and enables improved recyclability through accurate monitoring of materials [16]. Returnable recycling systems using IoT offer a critical step towards a circular economy, reducing the need for new materials and improving the use of available resources. Despite advances in IoT technologies for food packaging, research gaps remain, particularly in terms of integrating these technologies into returnable recycling systems. The lack of common standards and data security remain challenges that need to be addressed in future research.

This study examines the contribution of new packaging technologies incorporating IoT, focusing on their applications to improve sustainability in the food industry. Smart and active packaging, recyclable materials, and returnable recycling systems are analyzed, and the importance of traceability and real-time product tracking via IoT is also discussed. Recent research in 2024 highlights the importance of appropriate plans and guidelines to improve sustainability [17].

The use of IoT technologies in food packaging has similarities with other sectors, such as pharmaceuticals and automotives, where traceability is critical. However, food packaging requires more stringent standards given the nature of the products, and IoT technologies provide new opportunities to improve quality and safety. 

The use of IT in several areas of the food industry is the objective of many research initiatives. New packaging technologies, such as biodegradable materials and recyclable solutions, are instrumental in reducing the environmental footprint of the food industry. Active packaging, which uses IoT sensors to monitor product storage conditions, helps to ensure food quality and reduce waste [18]. New research from 2024 highlights the need to improve dairy packaging design to reduce waste by enhancing technical evacuation of products [5]. Recyclable packaging enables improved waste management and enhances material reuse practices, thus reducing pressure on environmental systems [7]. At the same time, the integration of IoT sensors and blockchain technologies offers opportunities to monitor products throughout the supply chain, promoting sustainability [16].

Moreover, there are also many studies about traceability and packaging technologies. Product traceability has become essential to ensure quality, safety, and consumer confidence. The integration of IoT technologies, such as RFID sensors and blockchain tags, allows for the continuous tracking of products from the production stage to final consumption [19]. In this way, consumers can be confident about the origin and quality of products, while facilitating returnable recycling and reuse of packaging [20]. The recyclability and use of returnable materials in food packaging is a critical element in reducing the environmental footprint. Packaging containing IoT sensors provides waste-tracking capabilities, improving recycling management and reducing the need for new materials [21]. Recent research in 2024 highlights that properly informing consumers about the safety of reusable packaging can increase their preference for these solutions, despite production costs [17]. Finally, there are many studies about the connection of IoT to the circular economy in food packaging. The application of IoT technologies in the circular economy of food packaging helps to improve recyclability and reduce resource waste. Real-time monitoring of packaging using IoT sensors enhances the efficiency of recycling processes, while the data generated from these packages help to improve resource and waste management processes [8].

Overall, the existing literature has highlighted the importance of sustainable packaging technologies and the integration of IoT in waste management. However, previous studies have not adequately explored the combination of IoT technologies with recycling solutions, leaving a gap in the literature [13,14]. This study aims to fill this gap.

### 2.2. Bibliometric Analysis

The objective of this part of the study is to identify the common themes related to IoT and food packaging examined in the literature. A bibliometric analysis was conducted, and the Scopus, Web of Science and IEEE Xplore databases were used to perform the analysis. These databases offer a wide range of research articles covering scientific fields such as engineering, computer science, environmental science, and food science, thus providing a strong basis for the analysis of new packaging and IoT technologies.

The search strategy was based on the use of keywords related to IoT technologies and food packaging, such as “sustainable food packaging”, “IoT-enabled packaging”, “recyclable packaging technologies”, and “food supply chain traceability”. The search was conducted to cover all relevant research articles published from 2010 to 2024. The articles included in the study met the following criteria: publications in English, exploration of packaging technologies with a focus on IoT, investigation of solutions to improve the sustainability and recyclability of packaging, and the presence of empirical data or methods of bibliometric analysis. Data analysis was performed using VOS viewer, which allows the creation of thematic maps and keyword co-occurrence analysis. Through this process, the relationships between IoT technologies and sustainability in food packaging were highlighted, while key geographical and institutional trends were identified [22].

The bibliometric analysis revealed that publications regarding IoT packaging technologies have increased significantly since 2015, reinforcing the emphasis on sustainability. The main keywords used included the terms “circular economy”, “smart packaging”, “recyclable packaging” and “sustainable packaging”, highlighting the link between these technologies and the circular economy and waste management.

The following analysis presents two visual representations (Figure 1 and Figure 2) created with the VOS viewer tool. These images highlight the thematic areas and the connections between them based on the keywords used in the research on IoT and food packaging.

The first figure shows the connection between keywords such as “Internet of Things (IoT)”, “traceability”, “circular economy” and “food supply chains”. We can see that IoT technologies are directly linked to “precision agriculture”, “food supply chains” and “food safety”, enhancing sustainability in food production and distribution. The keywords linked to “Industry 4.0” and “big data analytics” also demonstrate the significant contribution of new technologies to process monitoring and resource and waste management in the food industry.

The second figure, in heatmap format, reveals the density of publications and subject areas examined. The area around the “Internet of Things” appears as the most densely researched area, with IoT technologies having a direct link to developments in agriculture, food production, and supply chains. The areas of ‘traceability’ and ‘recycling systems’ have also attracted great interest in the study of sustainability through new technologies. This mapping reinforces the correlation of key areas of IoT application in food packaging with other technologies aimed at reducing waste and improving the efficiency of supply chains.

To further highlight the applications of new technologies in food packaging, we present the following table that categorizes the different application and technology areas related to IoT and sustainability (no direct relationship to our study) (Table 1). 

The above table incorporates the key application areas and evolving trends related to agriculture and agribusiness, as well as the challenges and opportunities presented in the context of Industry 4.0 and data analytics, with a focus on food sustainability and safety. The US, China, and E.U. countries such as Germany and Italy emerged as the key sources of scientific research and innovation in the field of sustainable packaging and IoT technologies. Research teams in these regions are developing significant collaborations to drive innovative solutions aimed at improving sustainability and resource management.

Overall, the findings of the bibliometric analysis on IoT and food packaging can be summarized as follows:Current applications of smart and active packaging in the food industry can enhance freshness, ensuring the traceability and efficiency of food safety monitoring systems.Recyclable materials and returnable recycling systems are proving effective in reducing waste and promoting the circular economy.Traceability and real-time product tracking via IoT significantly improve supply chain management and food safety.The main barriers for the intention to use IoT technologies include lack of standardization, security concerns, and implementation costs. However, in addition to the barriers, the factors that affect the behavioral intention of IoT regarding the use of IoT should also be identified.

### 2.3. Research Methodology

This part of the research aims to identify the factors that affect behavioral intention regarding the use of Internet of Things (IoT). More specifically, we examine (a) the effect of perceived usefulness, ease of use, self-efficacy, and personal innovativeness, on, performance expectancy from IoT and attitudes towards IoT, and (b) the effect of performance expectancy and attitude towards IoT on the behavioral intention to use IoT in food packaging in the future. The respondents were managers of companies in the food industry. 

The research questionnaire is based on the Technology Acceptance Model (TAM), which is one of the most successful models for understanding technology acceptance behavior [23]. The main goal of TAM is to provide a framework for exploring the effects of external factors on internal views, behaviors, and intentions. In this study, the TAM model is integrated with the Macik and Curie-Sklodowska (published at 2017) model. The factors examined are presented in the following conceptual model (Figure 3).

Usefulness is a key variable related to technology acceptance from a consumer perspective and can be defined as the “extent to which a person’s performance improves when using a new technology” [23,24]. Ease of use refers to “the extent to which a person believes that using a particular system is effortless” [25]. Self-efficacy refers to “the performance of activities to produce results with a given capacity by a product, service, or entity” [26]. Personal innovativeness has been defined by Agarwal and Prasad [27] as “an individual’s willingness to try a new information technology”. Performance expectancy refers to “the extent to which a person believes that using the system will help them increase their job performance or the extent to which the technology will provide benefits to the consumer” [28]. Attitude is a person’s evaluation of a particular behavior and is defined as “a person’s feelings (positive or negative) about performing the target behavior and is a key factor in the acceptance and use of information technology” [29]. Behavioral intention is also defined as “customers’ tendency and willingness to adopt new technology and is an important predictor of technology use or a fundamental factor in determining individual behavior toward new technologies” [23,30].

Therefore, the following research hypotheses are proposed:

**H_1_**: 
*Perceived usefulness, ease of use, self-efficacy, and personal innovativeness affect positively performance expectancy from IoT and attitude towards IoT.*


**H_2_**: 
*Performance expectancy and attitude towards IoT affect positively the behavioral intention to use IoT in the future.*


A suitable research tool was employed to gather relevant data to test the research hypotheses and the conceptual model. Since this is primary research with a focus on the attitudes and intentions of potential technology users, an exploratory survey was conducted with the use of a closed type of questionnaire. This practice was selected because questionnaires enhance the consistency and reliability of the research findings and facilitate the collection of adequate data within a limited period [31].

The questionnaire was divided into two sections; the first included demographic questions, collecting information such as gender, age, education, years of experience, and employment position. The second part of the questionnaire comprised closed-ended questions adapted from established instruments aimed at assessing factors like usefulness, ease of use, self-efficacy, personal innovativeness, performance expectancy, attitude, and behavioral intention. Table 2 provides an overview of the questionnaire items. Responses were measured using a Likert scale ranging from “1 = strongly disagree” to “5 = strongly agree.” The 5-point Likert scale was used, as it is less confusing and increases the rate and quality of responses [32,33].

For instance, in the case of perceived usefulness, values close to ‘1’ reflect low perceived usefulness of the technology, while values near ‘5’ indicate high perceived usefulness.

The questionnaire was made available online via Google Forms and was distributed through email to several potential participants working in the food industry. Each email contained a link to the questionnaire. Additionally, the link was spread in several social media groups and networks. Convenience sampling, a non-probability sampling method, was used. Since participants needed access to email or the internet, individuals without internet access were naturally excluded from the population frame. However, due to the nature of this research, which requires at least basic internet and technology knowledge to accurately measure personal attitudes towards technology, this sampling method is not expected to introduce significant bias. It is anticipated to produce a reasonably representative sample. Finally, regression analysis was used to address the research questions and align them with the conceptual model as it is considered a powerful and flexible tool for data analysis [34].

## 3. Results

A total of 162 usable questionnaires were collected. Participants were senior managers in the food industry who have the knowledge to provide meaningful insights into the food packaging paradigm. Their responses were analyzed to obtain empirical evidence for the two research hypotheses. The demographic data are shown in Table 3.

There is relatively even representation of both genders in the survey, the age distribution of the respondents is concentrated in the range of 41–50 (53.6%), with 62.44% of the respondents belonging to this category. The next largest age groups are 31–40 and 51–65, both covering 30% of respondents. The largest percentages of the sample are BSc and MSc degree holders (43.6% and 49.5 % respectively). Regarding employment position, most of the sample (68.7%) are employed in operations, while 22.7% are in marketing and sales, and only 8.6% are working in finance and accounting. Finally, approximately three out of four respondents have more than 6 years of working experience.

In the Table 4 below, all the constructs present a high range of reliabilities. More specifically, all constructs exhibit Cronbach’s alpha values above the 0.70 threshold, thus suggesting good internal consistency.

### 3.1. Performance Expectancy from IoT

In Table 5, the results of the regression analysis for performance expectancy from IoT as a dependent variable (research hypothesis H_1_) are presented.

The F-statistic values show that the regression equation is overall statistically significant at the 0.1% level (F = 16.32, *p* < 0.001), indicating that at least one independent variable affects the dependent. Based on the adjusted-R^2^ value, the variability of performance expectancy from IoT can be explained by 38.6% from independent variables variability, implying a middling goodness of fit of the data on this regression. Therefore, given the statistically significant regression and the middling extent of explanatory power, it is reasonable to test each coefficient’s statistical significance separately to provide evidence about each independent variable’s effect, according to research hypotheses. The conceptual model and findings are shown in Figure 4.

### 3.2. Attitude Towards IoT

In Table 6, the results of the regression analysis for attitude towards IoT as a dependent variable (research hypothesis H_1_) are presented. 

The F-statistic values show that the regression equation is overall statistically significant at 0.1% level (F = 15.51, *p* < 0.001), indicating that at least one independent variable affects the dependent. According to the adjusted R2, the variability of attitude towards IoT can be explained by the 37.5% variability in the independent variables, implying a middling goodness of fit of the data on this regression. Therefore, given the statistically significant regression and the middling extent of explanatory power, it is reasonable to test each coefficient’s statistical significance separately to provide evidence about each independent variable’s effect, according to the research hypotheses. The suggested model and findings are shown in Figure 5. 

### 3.3. Behavioral Intention: Future Use of IoT

In Table 7, the regression results concerning research hypothesis H_2_ are presented.

The F-statistic values show that the regression equation is overall statistically significant at 0.1% level (F = 73.28, *p* < 0.001), indicating that at least one independent variable affects the dependent. According to the adjusted R2, the variability of future use of IoT can be explained the 62.3% variability in the independent variables, implying a high goodness of fit of the data on this regression. Therefore, given the statistically significant regression and the middling extent of explanatory power, it is reasonable to test each coefficient’s statistical significance separately to provide evidence about each independent variable’s effect, according to the research hypotheses. The suggested model and the results are presented in Figure 6. 

Therefore, regarding research hypothesis H_1_, the idea that the positive effect of personal innovativeness on attitude toward IoT becomes stronger when there is awareness of IoT is supported. However, this is not the case for usefulness, ease of use, and self-efficacy. More specifically, the positive effect of perceived usefulness strengthens with IoT awareness, while the non-significant effect of ease of use remains largely unchanged, regardless of IoT awareness. In contrast, the positive effect of self-efficacy weakens and becomes non-significant when there is awareness of IoT. Regarding research hypothesis H_2_, the positive effect of attitude toward IoT on the intention to use IoT in the future becomes stronger when there is awareness of IoT. However, this is not the case for performance expectancy from IoT. Specifically, the positive impact of performance expectancy on future IoT use remains nearly the same, regardless of whether there is awareness of IoT or not.

## 4. Conclusions

This study contributes to the discussion on the potential use of IoT in food packaging. The primary objective of the first part of the study was to identify common themes related to IoT and food packaging as examined in the literature. The thematic–bibliometric analysis on IoT and food packaging revealed several important findings.

The current applications of smart and active packaging in the food industry enhance freshness, ensure traceability, and improve the efficiency of food safety monitoring systems. Similarly, past research has underlined the importance of IoT in monitoring freshness, traceability, and food safety [35,36]. Recyclable materials and returnable recycling systems have proven to be effective solutions in reducing waste and promoting a circular economy. Likewise, previous research confirms that implementing IoT technologies in the food supply chain can significantly reduce food waste through improved real-time monitoring and management of products, while simultaneously enhancing food traceability and safety [37,38,39].

Furthermore, IoT-enabled traceability and real-time product tracking significantly improve supply chain management and food safety. These findings are consistent with other research, which also indicates that IoT applications substantially increase the efficiency of food supply chain management and traceability [40,41,42]. However, the main barriers to the adoption of IoT technologies include a lack of standardization, data security concerns, and high implementation costs. Similar barriers to IoT adoption in the food industry have also been observed in studies focusing on developing countries [43]. This similarity is particularly interesting because Greece is classified as an advanced economy in the IMF World Economic Outlook database (2024) [44], where differences between developing and advanced economies might be expected. This finding may partly reflect the fact that Greece’s digital literacy rate remains below the EU average [45].

The findings of this research underscore the critical role of IoT technologies in enhancing sustainability and efficiency in food packaging. The integration of IoT technologies presents significant opportunities, particularly through active and smart packaging, recyclable solutions, and returnable recycling systems. These innovations offer considerable benefits in reducing waste and promoting a circular economy. Companies adopting these technologies can minimize their environmental footprint and optimize resource management. Moreover, IoT-enabled tracking systems enhance traceability and facilitate improved product quality management.

However, the successful application of these technologies necessitates careful planning and an in-depth understanding of the factors influencing their adoption. A key finding of this analysis was the identification of barriers to IoT adoption. Nonetheless, the factors influencing IoT adoption were not thoroughly examined. The primary research conducted in this study seeks to address this gap by identifying the factors affecting behavioral intention toward the future use of IoT. A combination of the TAM model and the Macik and Curie-Sklodowska (2017) model is employed, with a specific focus on the determinants influencing behavioral intention to adopt IoT in the food industry.

The findings from this primary research demonstrate the positive impact of (1) perceived usefulness, (2) ease of use, (3) self-efficacy, and (4) personal innovativeness on both (a) performance expectancy of IoT and (b) attitudes toward IoT. Furthermore, the results indicate that performance expectancy and attitudes toward IoT positively influence the behavioral intention to adopt IoT in the future. This research highlights the vast potential of IoT for the food industry [46].

These findings are particularly valuable for companies aiming to enhance their capacity for IoT adoption. IoT has the potential to revolutionize the food industry and address various challenges facing the food supply chain. IoT applications can improve food product quality and freshness, enhance food safety, and increase the efficiency of food packaging recycling. However, barriers to wider adoption remain, including a lack of standardization, data security concerns, and implementation costs. These findings emphasize the importance of understanding the factors influencing IoT adoption in the food industry. Companies seeking to integrate IoT technologies into food packaging should prioritize demonstrating the utility and ease of use of these technologies while fostering user self-efficacy and innovation. Sharing best practices among industry stakeholders is also crucial [12,47,48,49].

Additionally, firms within the food supply chain should invest more in the integration of digital technologies. At the current rate, the gap between Greece and other EU countries is unlikely to narrow [50].

The study does have limitations, including the use of convenience sampling for the survey and potential biases arising from self-reported data. Future research could extend the model by incorporating additional factors such as data security and environmental awareness, as well as exploring differences across various sectors of the food industry. Moreover, further research is needed to improve the integration of IoT technologies into existing waste management and recycling systems. Lastly, the same approach could be applied to assess IoT adaptability in other industries.

## Figures and Tables

**Figure 1 foods-14-00575-f001:**
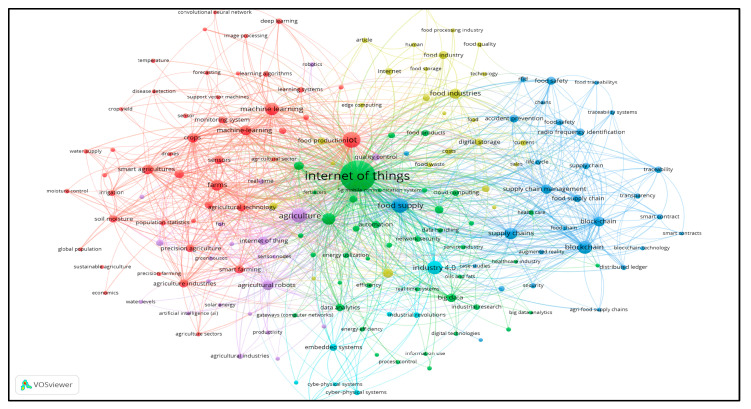
Network map.

**Figure 2 foods-14-00575-f002:**
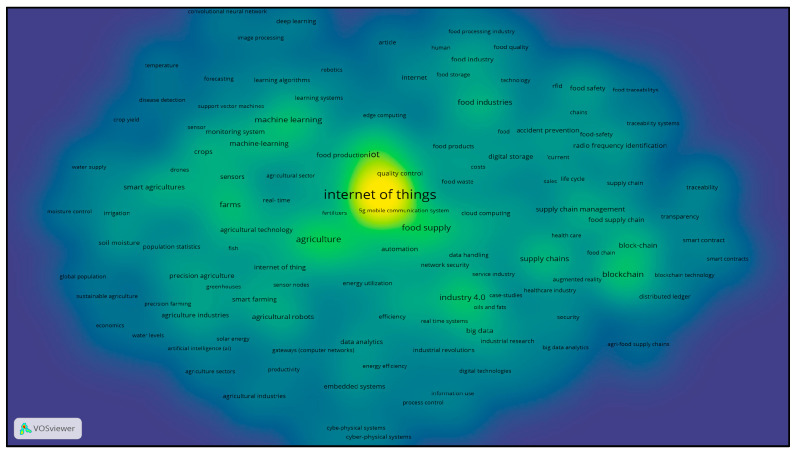
Density map.

**Figure 3 foods-14-00575-f003:**
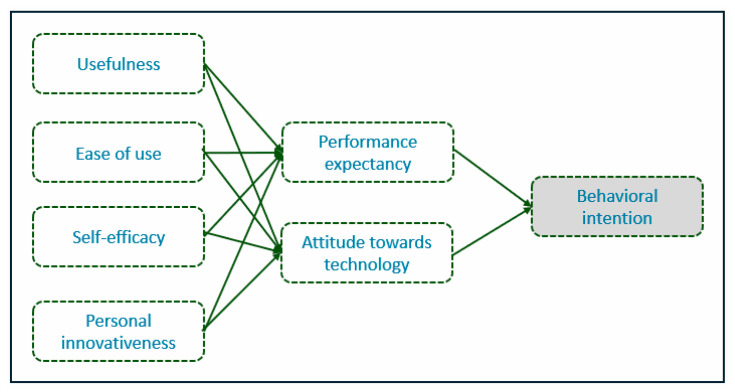
Conceptual TAM and Macik and Curie-Sklodowska model combination.

**Figure 4 foods-14-00575-f004:**
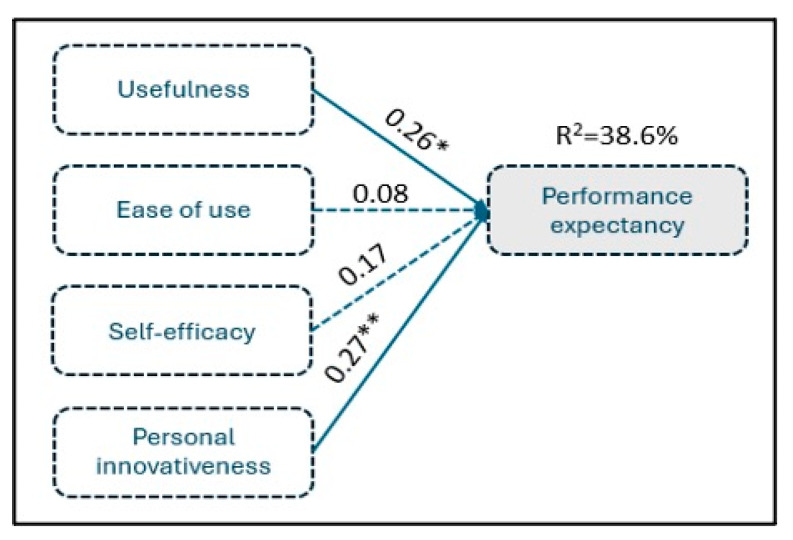
Conceptual model estimation for performance expectancy—research hypothesis H_1._ * Statistically significant at 5%; ** statistically significant at 1%.

**Figure 5 foods-14-00575-f005:**
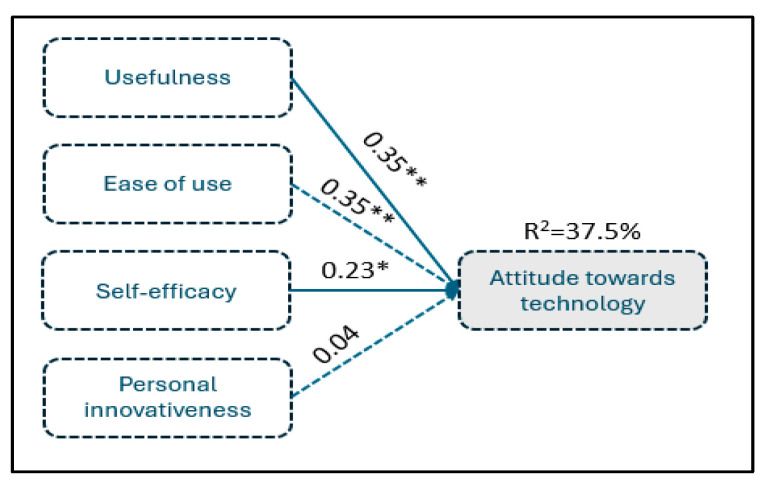
Conceptual model estimation for attitude towards technology—research hypothesis H_1_. * Statistically significant at 5% **; statistically significant at 1%.

**Figure 6 foods-14-00575-f006:**
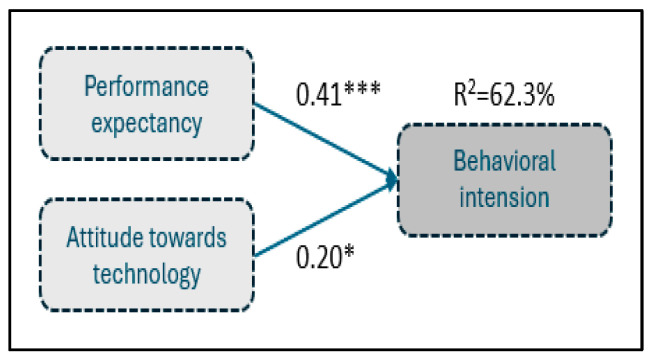
Conceptual model estimation for behavioral intension—research hypothesis H_2_. * Statistically significant at 5%; *** statistically significant at 0.1%.

**Table 1 foods-14-00575-t001:** IT applications related to IoT and sustainability.

Areas	Themes
Application Domains	Agricultural industries, agri-food supply chains, agricultural sector, food processing Industry, animals, aquaculture, developing countries, farming system, farms, food traceability, water management.
Industry	Cost-effectiveness, crop yield, cultivation, global population, greenhouses, population statistics, precision agriculture, precision farming, smart farming, sustainable agriculture, transparency.
Evolving Paradigms	Agricultural robots, automation, convolutional neural network, cyber–physical systems, drones, intelligent systems, interoperability, microcontrollers, moisture control, monitoring system, network architecture, network security, smart agriculture, soil moisture, support vector machines.
Industry 4.0	Agriculture 4.0, 5g communication systems, artificial intelligence, augmented reality, bid data analytics, blockchain technology, cloud computing, embedded systems, radio frequency identification (rfid), robotics, sensor networks, sensor nodes.
Data Analysis	Accident prevention, data acquisition, data analytics, data handling, decision making, decision support systems, decision trees, deep learning, digital storage, digital technologies, digital transformation, forecasting, learning algorithms, learning systems, machine learning.
Food Safety	Climate change, disease detection, energy efficiency, energy utilization, environmental impact, fertilizers, food quality, food safety, food storage, waste management.

**Table 2 foods-14-00575-t002:** TAM factors and the corresponding questions.

Factors	Questions/Items
Usefulness	Tasks would be difficult without IoT. Using IoT gives me greater control over the execution of the tasks. Using IoT improves staff performance. Using IoT saves staff time. Using IoT allows staff to accomplish tasks more quickly. Using IoT allows staff to accomplish more things than would be possible. Using IoT reduces the time staff spends on unproductive activities. Using IoT enhances effectiveness. Using IoT increases productivity. Overall, I find IoT to be useful.
Ease of use	Staff often become confused when they use IoT. Staff make errors frequently when using IoT. Interacting with IoT is often frustrating. Staff need to consult the user manual often when using IoT. Interacting with IoT requires a lot of mental effort. Staff find it easy to recover from errors encountered when using IoT. Staff find it easy to get IoT to do what they want it to do. IoT often behaves in unexpected ways. Staff find it cumbersome to use IoT. Staff interaction with IoT is easy for me to understand. It is easy for staff to remember how to perform tasks using IoT.
Self-efficacy	If something looks too complicated, staff will not even bother to try it. Staff avoid trying to learn new things when they look too difficult. When trying to learn something new, staff soon give up if they are not initially successful. When staff members make plans, they are certain I can make them work. If they can’t do a job the first time, they keep trying until they can. When staff members have something unpleasant to do, I stick to it until I finish it. When they decide to do something, they can go right to work on it. Failure just makes staff try harder. When staff set important goals for themselves, they rarely achieve them. Staff don’t seem capable of dealing with most problems that come up in the execution of their tasks. When unexpected problems occur, staff don’t handle them very well. Staff feel insecure about my ability to do things.
Personal innovativeness	If staff become aware of a new information technology, they would look for ways to experiment with it. In general, staff members are hesitant to try out new information technologies. They like to experiment with new information technologies.
Performance expectancy	Staff find the use of IoT and its devices useful in their everyday life at work. Using IoT and its devices increases staff chances of achieving the objectives important to them. Using IoT and its devices allows staff to live more comfortably. Using IoT and its devices leads to a more productive use of their time.
Attitude towards IoT	It would be a wonderful idea to use IoT. Staff would have positive feelings toward IoT. It is easier and better for staff to use IoT compared to traditional technologies.
Future use of IoT	Staff intend to continue in the future using IoT and its devices. Staff try to use the IoT and its devices as often as they can. They are going to continue to use IoT and its devices frequently. They are going to continue to use IoT and its devices in the future.

**Table 3 foods-14-00575-t003:** Sample demographics.

*Gender*	N	%
Male	84	51.4
Female	79	48.6
*Age group*		
18–30	24	14.5
31–40	24	15.0
41–50	87	53.6
51–65	25	15.5
Over 65	2	1.4
*Educational level*		
Up to high school	11	6.9
BSc/College	71	43.6
MSc/PhD	81	49.5
Employment Position		
Operations	112	68.7
Marketing and Sales	37	22.7
Finance and Accounting	14	8.6
*Years of experience*		
1–5	45	27.6
6–15	93	57.1
More than 15	25	15.3

**Table 4 foods-14-00575-t004:** Reliability of constructs.

Scale	Items	Cronbach α
Usefulness	10	0.839
Ease of Use	11	0.805
Self-Efficacy	12	0.867
Personal Innovativeness	3	0.811
Performance Expectancy from IoT	4	0.890
Attitude towards IoT	3	0.858
Future Use of IoT	4	0.889

**Table 5 foods-14-00575-t005:** Regression analysis between performance expectancy and usefulness, ease of use, self-efficacy and personal innovativeness.

Regression Coefficients
The constant term is 0.43, with a t-value of 0.78 and a *p*-value of 0.4378. This indicates that the constant is not statistically significant at any conventional level (such as 0.05 or 0.01). The coefficient for usefulness is 0.26, with a t-value of 2.09 and a *p*-value of 0.0187. This indicates that it is positively and significantly related to the dependent variable, performance expectancy, at the 0.05 level. The coefficient for ease of use is 0.08, with a t-value of 0.64 and a *p*-value of 0.2609. This indicates that it is not significantly related to performance expectancy. The coefficient for self-efficacy is 0.17, with a t-value of 1.25 and a *p*-value of 0.1072. This indicates that it is positively and significantly related to performance expectancy. The coefficient for personal innovativeness is 0.27, with a t-value of 2.38 and a *p*-value of 0.0092. This indicates that it is positively and significantly related to the dependent variable, performance expectancy, at the 0.01 level.

**Table 6 foods-14-00575-t006:** Regression analysis between attitude towards technology and usefulness, ease of use, self-efficacy and personal innovativeness.

Regression Coefficients
The constant term is 0.38, with a t-value of 0.70 and a *p*-value of 0.4865. This indicates that the constant is not statistically significant at any conventional level (such as 0.05 or 0.01). The coefficient for usefulness is 0.35, with a t-value of 2.88 and a *p*-value of 0.0022. This indicates that it is positively and significantly related to the dependent variable, attitude towards IoT, at the 0.01 level. The coefficient for ease of use is 0.18, with a t-value of 1.59 and a *p*-value of 0.0567. This indicates that it is not significantly related to attitude towards IoT. The coefficient for self-efficacy is 0.23, with a t-value of 1.73 and a *p*-value of 0.0428. This indicates that it is positively and significantly related to attitude towards IoT. The coefficient for personal innovativeness is 0.04, with a t-value of 0.39 and a *p*-value of 0.3471. This indicates that it is not significantly related to attitude towards IoT.

**Table 7 foods-14-00575-t007:** Regression analysis between behavioral intention and future use of IOT with performance expectancy and attitude towards IoT.

Regression Coefficients
The constant term is 1.26, with a t- value of 4.43 and a *p*-value of 0.000. This indicates that the constant is statistically significant at any conventional level (such as 0.05 or 0.01). The coefficient for performance expectancy from IoT is 0.41, with a t-value of 4.25 and a *p*-value of 0.0000. This indicates that it is positively and significantly related to the dependent variable, attitude towards IoT, at the 0.001 level. The coefficient for attitude towards IoT is 0.20, with a t-value of 1.92 and a *p*-value of 0.0284. This indicates that it is positively and significantly related to the dependent variable, attitude towards IoT, at the 0.05 level.

## Data Availability

The original contributions presented in the study are included in the article, further inquiries can be directed to the corresponding author.

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
