# Peer review of "Exploring the Factors Influencing the Acceptance of IoT Applications in Food Packaging"

_foods, 2025, doi:10.3390/foods14040575_

Round 1
Reviewer 1 Report
Comments and Suggestions for Authors
The authors have presented an overview of the application of IoT in food packaging. They have addressed this in two ways. The first is through a bibliographic overview of the scientific publications in the last 14 years and the second is through a questionaire.
Though these subjects have been mostly adequately addressed and explained the paper lacks what the title promises to give - trends, prospects and challenges in terms of the actual IoT solutions that are used.
The authors also need to address the trends, prospects and challenges of IoT (smart packaging solutions) also through a review of what is written in the publications in the last 14 years.
Table 1 would be better given as text.
Besides the performed bibliographic overview the paper needs to give an overview of the smart solutions that are applied. This is lacking. The authors address keywords and give some maps, but there is no information or details on the actual trends and future prospects of IoT use in food packaging.
The questionaire is well thought out and the discussion is good.
The conclusion is not based on the authors paper, but is based on literature and this needs to be changed. Each paragraph refers to one paper/reference. All discussion needs to be in the paper and then the conclusion needs to be based on the what was said/given in the paper.
Author Response
The authors have presented an overview of the application of IoT in food packaging. They have addressed this in two ways. The first is through a bibliographic overview of the scientific publications in the last 14 years and the second is through a questionnaire.
Comment 1: Though these subjects have been mostly adequately addressed and explained, the paper lacks what the title promises to give - trends, prospects and challenges in terms of the actual IoT solutions that are used.
Response 1: Thank you for your comment. The primary objective of our research was to identify the factors that influence the acceptance of IoT applications in food packaging, which was also the focus of our primary data analysis. In light of this, we have revised the title to: 'Exploring the Factors Influencing the Acceptance of IoT Applications in Food Packaging,' and have updated the entire paper accordingly.
Comment 2: The authors also need to address the trends, prospects and challenges of IoT (smart packaging solutions) also through a review of what is written in the publications in the last 14 years.
Response 2: You are correct. The literature review focuses on the integration of IoT technologies in food packaging and the potential use of IoT applications within the targeted industry.
Comment 3: Table 1 would be better given as text.
Response 3: We have integrated the paper's content into the text and renumbered the tables accordingly.
Comment 4: Besides the performed bibliographic overview the paper needs to give an overview of the smart solutions that are applied. This is lacking. The authors address keywords and give some maps, but there is no information or details on the actual trends and future prospects of IoT use in food packaging.
Response 4: You are correct, and with the changes we have made, we believe we have effectively focused on the paper's main objective.
Comment 5: The questionnaire is well thought out and the discussion is good.
Response 5: Thank you for your comment; it is truly appreciated.
Comment 6: The conclusion is not based on the authors paper but is based on literature and this needs to be changed. Each paragraph refers to one paper/reference. All discussion needs to be on paper and then the conclusion needs to be based on what was said/given in the paper.
Response 6: Thank you for this comment. We have revised the opening paragraphs of the Conclusion section to provide a more coherent analysis of our research findings while also comparing them with similar studies.
Reviewer 2 Report
Comments and Suggestions for Authors
This article provides a more systematic overview of the transformative role of IoT in improving efficiency across the food value chain. It combines secondary and primary research to identify trends, challenges and prospects for IoT adoption in the food industry. The paper is more systematic and logical, responding to the research frontiers and issues, especially the in-depth study of the factors affecting IoT adoption, which is useful for research in the field, but still has the following issues:
1. Fewer references only 42, and the proportion of literature in the last three years is low, which does not cover some of the latest research results in the field, it is recommended to supplement the latest high-quality literature to provide a more complete review.
2. There are fewer references in the latter part of the article and most of them are references already cited in the previous content, please add new references to support the ideas described.
3. Focus the review on the core content, and reduce or omit unnecessary statements or content that is not related to the content of the study.
4. You can embellish the formatting of the tables to make them clearer and easier to read.
5. Unify the format and size of fonts in Figure.
6. The conclusion section is too busy, please streamline the language and highly summarize based on the core content.
7. Pay attention to errors in formatting, symbols, and font syntax in the text, read the article carefully and revise it.
Author Response
This article provides a more systematic overview of the transformative role of IoT in improving efficiency across the food value chain. It combines secondary and primary research to identify trends, challenges and prospects for IoT adoption in the food industry. The paper is more systematic and logical, responding to the research frontiers and issues, especially the in-depth study of the factors affecting IoT adoption, which is useful for research in the field, but still has the following issues:
Comment 1: Fewer references only 42, and the proportion of literature in the last three years is low, which does not cover some of the latest research results in the field, it is recommended to supplement the latest high-quality literature to provide a more complete review.
Response 1: We have enhanced the literature review by incorporating eight updated references that address IoT technologies and the Technology Acceptance Model (TAM) application. These additions provide a more comprehensive perspective on the current advancements and research in these areas, ensuring the review is thorough and up-to-date.
Comment 2: There are fewer references in the latter part of the article and most of them are references already cited in the previous content, please add new references to support the ideas described.
Response 2: Done – see our above response.
Comment 3: Focus the review on the core content and reduce or omit unnecessary statements or content that is not related to the content of the study.
Response 3: Done.
Comment 4: You can embellish the formatting of the tables to make them clearer and easier to read.
Response 4: Done. We have integrated the paper's content into the text and renumbered the tables accordingly.
Comment 5: Unify the format and size of fonts in Figure.
Response 5: Done.
Comment 6: The conclusion section is too busy, please streamline the language and highly summarize based on the core content.
Response 6: Thank you for this comment. We have revised the opening paragraphs of the Conclusion section to provide a more coherent analysis of our research findings while also comparing them with similar studies.
Comment 7: Pay attention to errors in formatting, symbols, and font syntax in the text, read the article carefully and revise it.
Response 7: Done.
Round 2
Reviewer 1 Report
Comments and Suggestions for Authors
The authors have answered all the questions raised in the review and have implemented the required changes in their revised paper. Therefore I recommend this work for publication in its present form.
Reviewer 2 Report
Comments and Suggestions for Authors
accept